# Intrinsic and Extrinsic Control of Hepatocellular Carcinoma by TAM Receptors

**DOI:** 10.3390/cancers13215448

**Published:** 2021-10-29

**Authors:** Viola Hedrich, Kristina Breitenecker, Leila Djerlek, Gregor Ortmayr, Wolfgang Mikulits

**Affiliations:** Comprehensive Cancer Center, Department of Medicine I, Institute of Cancer Research, Medical University of Vienna, 1090 Vienna, Austria; viola.hedrich@meduniwien.ac.at (V.H.); kristina.breitenecker@meduniwien.ac.at (K.B.); a01306912@unet.univie.ac.at (L.D.); gregor.ortmayr@meduniwien.ac.at (G.O.)

**Keywords:** hepatocellular carcinoma, receptor tyrosine kinase, TAM, Tyro3, Axl, MerTK, tumor microenvironment, chemoresistance, therapy

## Abstract

**Simple Summary:**

Tyro3, Axl, and MerTK are receptor tyrosine kinases of the TAM family, which are activated by their ligands Gas6 and Protein S. TAM receptors have large physiological implications, including the removal of dead cells, activation of immune cells, and prevention of bleeding. In the last decade, TAM receptors have been suggested to play a relevant role in liver fibrogenesis and the development of hepatocellular carcinoma. The understanding of TAM receptor functions in tumor cells and their cellular microenvironment is of utmost importance to advances in novel therapeutic strategies that conquer chronic liver disease including hepatocellular carcinoma.

**Abstract:**

Hepatocellular carcinoma (HCC) is the major subtype of liver cancer, showing high mortality of patients due to limited therapeutic options at advanced stages of disease. The receptor tyrosine kinases Tyro3, Axl and MerTK—belonging to the TAM family—exert a large impact on various aspects of cancer biology. Binding of the ligands Gas6 or Protein S activates TAM receptors causing homophilic dimerization and heterophilic interactions with other receptors to modulate effector functions. In this context, TAM receptors are major regulators of anti-inflammatory responses and vessel integrity, including platelet aggregation as well as resistance to chemotherapy. In this review, we discuss the relevance of TAM receptors in the intrinsic control of HCC progression by modulating epithelial cell plasticity and by promoting metastatic traits of neoplastic hepatocytes. Depending on different etiologies of HCC, we further describe the overt role of TAM receptors in the extrinsic control of HCC progression by focusing on immune cell infiltration and fibrogenesis. Additionally, we assess TAM receptor functions in the chemoresistance against clinically used tyrosine kinase inhibitors and immune checkpoint blockade in HCC progression. We finally address the question of whether inhibition of TAM receptors can be envisaged for novel therapeutic strategies in HCC.

## 1. Hepatocellular Carcinoma

Hepatocellular carcinoma (HCC) is one of the most common malignancies worldwide and accounts for 75–85% of liver cancer patients [1]. The mortality rate has increased by more than 50% in the last two decades, and it is estimated that over 1 million people will be diagnosed with liver cancer in 2025 due to rising incidence [2]. The most relevant risk factors for HCC are chronic infection with hepatitis B or C virus (HBV, HCV), diabetes- or obesity-related non-alcoholic fatty liver disease (NAFLD)/non-alcoholic steatohepatitis (NASH), and permanent alcohol abuse, which lead to premalignant liver cirrhosis [3]. The global distribution of HCC is diverse, as geographically dependent risk factors correlate with the development of the disease. Etiological factors such as NASH, HCV infection, and alcohol intoxication play a predominant role in the Western hemisphere including the USA and Europe, while HBV/HCV infection and dietary exposure to aflatoxin are most prevalent risk factors in Eastern and Southern hemispheres, including Asia and Africa.

HCC is a heterogeneous disease at the molecular and cellular level by displaying variations based on the comparison between patients, tumors, and intra-tumoral nodules [4]. The interpatient heterogeneity has been determined by two major subclasses of HCC patients linked to proliferation and non-proliferation, which can be further subdivided by etiological factors, cell signaling, and chromosomal stability. The proliferation class largely associates with HBV infection and worse clinical outcomes, whereas the non-proliferation class is accompanied by HCV infection and alcohol intoxication, showing exclusion of immune cells [4]. With respect to inter-tumoral heterogeneity, tumor nodules can evolve either by de novo carcinogenesis, generating independent primary tumors, a process designated as multi-centric tumor occurrence (MO), or by dissemination of cancer cells via vessel invasion and exit from circulation, a process known as intrahepatic metastasis (IM). Recent studies focusing on MO and IM suggest that 35 % of HCC patients harbor IM-derived tumor nodules and that nodules formed by MO or IM display the same driver mutations [5]. While MO can be diagnosed at rather early stages, IM emerges at later stages of HCC progression. Patients with recurrent HCC due to IM and MO show better prognosis in the MO group with lower mortality and better disease-free survival compared to those in the IM group [6]. Intra-tumoral heterogeneity is based on activation of driver genes, changes in the epithelial plasticity of malignant hepatocytes by epithelial to mesenchymal transition (EMT), the role of cancer stem cells in HCC, and the tumor microenvironment [4]. Notably, intra-tumoral heterogeneity has a large clinical impact on the treatment of drug-resistant HCC. 

Staging of HCC patients is widely performed according to the Barcelona Clinic Liver Cancer Classification (BCLC), which quantifies tumor burden together with the extent of liver dysfunction [7]. Patients at early stages BCLC-0 or BCLC-A harboring a low number of small tumor nodules are candidates for surgical resection, while patients at early stages who are excluded from surgical resection are subjected to liver transplantation or ablation by, e.g., image-guided ethanol injection. Patients at intermediate-stage BCLC-B with large or multinodular HCC mainly receive transarterial chemoembolization, which combines chemotherapy with vessel plugging. HCC is frequently diagnosed at advanced stages, as only a low percentage of patients with chronic liver diseases (CLDs) are screened for HCC. Patients at advanced stage BCLC-C exhibiting multinodular HCC plus vascular invasion require systemic therapy by treatment with the tyrosine kinase inhibitors (TKIs) sorafenib or lenvatinib at first line and regorafenib or cabozantinib at second line [8,9]. In addition, HCC patients refractory to frontline treatments are treated with ramucirumab, targeting vascular endothelial growth factor (VEGF) receptor-2, or the immune checkpoint inhibitor nivolumab, targeting programmed cell death 1 (PD-1). Combinations of TKIs and ramucirumab improve survival when used as first-line or second-line treatment in patients with advanced HCC. Major disadvantages of this type of treatment are emerging adverse events and the limited efficacy due to primary or secondary resistance. Molecular therapies directed against novel targets such as transforming growth factor (TGF)-β, MET, and fibroblast growth factor (FGF) receptor 4 aim to better meet the biological tumor profile [10,11]. The combination of antiangiogenic drugs with immunotherapy shows therapeutic potential in the front-line setting. Current data highlight the importance of adopting novel combination strategies as HCC is not a “one drug disease” [11]. 

## 2. The Role of TAM Receptors in Cell Physiology

The receptor tyrosine kinases (RTKs) Tyro3, Axl, and MerTK constitute the TAM family. They were first identified in 1991 as novel protein tyrosine kinases [12] and were cloned as full length RTKs from central nervous system (Tyro3 [13]), chronic myelogenous leukemia (Axl [14]), and B-lymphoblastoid cells (MerTK [15]). In the 1990s, TAM receptors were de-orphanized by unveiling Gas6 and the serum factor Protein S (Pros1) as their high affinity ligands [16,17]. Gas6 strongly binds to Axl with a low picomolar equilibrium dissociation constant (Kd) [18] and with a low nanomolar Kd to MerTK and Tyro3, while Pros1 shows affinity to Tyro3 and MerTK but not to Axl [19]. TAM ligands are bridging molecules that bind with the amino-terminal domains to phosphatidylserine (PtdSer) residues externalized on membranes of apoptotic cells, aggregating platelets, or enveloped viruses, and by binding with the carboxy-terminus to TAM receptors [20]. Gas6 and Pros1 share the same domain organization, which is the N-terminal Gla region containing 11 c-carboxyglutamic acid residues, a loop region, four EGF-like repeats, and a C-terminal sex hormone-binding globulin (SHBG)-like structure consisting of two globular laminin G-like (LG) domains (Figure 1A). Less well characterized TAM ligands with low affinity to receptors are galectin 3 and tubby, which bind to MerTK, as well as tubby-like protein interacting with all TAMs [21]. Pros1 is highly abundant in the blood at 300 nM, while Gas6 is present at <0.2 nM and considered to be complexed with the soluble version of Axl (sAxl) [22,23]. 

Tyro3, Axl and MerTK localize on chromosome 15, 19, and 2, respectively, and consist of the N-terminal Ig1/Ig2 domains, two fibronectin (FN) domains, a transmembrane domain, and the cytoplasmic protein tyrosine kinase (PTK) domain (Figure 1A) [24]. Binding of Gas6 or Pros1 by the SHBG domain to the Ig1 region of TAM receptors forms a tetrameric 2:2 ligand–receptor complex after dimerization of the 1:1 ligand–receptor configuration, allowing RTK activation by phosphorylation of the cytoplasmic PTK domain. Homodimers of TAM receptors induce various signaling pathways including MAPK, Akt, Src, or PLCγ, which are involved in proliferation, survival, and cell invasion (Figure 1A). Variants of TAM receptors can affect their role in cell physiology. The Axl-short (Axl-S) splice variant, generated by PTBP1-induced skipping of exon 10, exhibits a more robust binding ability to Gas6 and stronger activation of Akt and Erk signaling, leading to a metastatic phenotype of HCC cells [25]. Development and progression of liver fibrosis in HCV-infected patients correlate with a single nucleotide polymorphism (SNP) in the MerTK locus (rs4374383) based on genome-wide association study data [26]. Patients with rs4374383 AA exhibited less frequently late stage fibrosis compared to those with rs4374383 GG/GA [26].

The Gla domain of Gas6 or Pros1 can be γ-carboxylated in a vitamin K-dependent fashion, which is required for the interaction of TAM receptors with PtdSer residues exposed at—membranes of activated platelets and apoptotic cells (Figure 1B) [27]. Binding of the γ-carboxylated Gla domain to PtdSer is the major feature of TAM receptors and provides the “eat-me” signal for efferocytosis, i.e., phagocytosis of apoptotic cells. The interaction of MerTK and Tyro3 with PtdSer-presenting membranes particularly enhances their receptor activation, suggesting a major impact on the clearance of apoptotic cells [20]. Efferocytosis mediated by TAM receptors is essentially involved in the removal of infiltrated immune cells during the resolution phase of inflammation. Thus, insufficient TAM activity can cause accumulation of necrotic cells and the release of self-antigens, resulting in the development of autoimmune disease. Accordingly, genetic loss of either Axl or MerTK or the combination of Axl and MerTK shows autoantibody production as well as defective clearance of apoptotic cells and enhanced levels of pro-inflammatory cytokines, leading to aggravated chronic inflammation (for a summary of TAM deficiencies, see [20]).

Importantly, the type I interferon (IFN)-α/β receptor (IFNAR) mediates an inflammatory response by inducing the expression of pro-inflammatory cytokines. However, activation of Toll-like receptors (TLRs) by pathogens induces the expression of inflammatory cytokines, such as type I interferons, which stimulate IFNAR to induce the signal transducer and activator of transcription (Stat)1-dependent expression of Gas6-Axl in dendritic cells and macrophages. Gas6/Axl physically interacting with IFNAR results in hybrid Gas6/Axl/IFNAR complexes and induces expression of the suppressor of cytokine signaling (SOCS)1/3, which inhibits janus kinases (JAKs) and allows inflammation to be curbed (Figure 1C) [28]. 

Maintenance of vessel integrity is crucially dependent on TAM receptor signaling. Gas6/Pros1 and TAM receptors are abundantly expressed by endothelial cells and vascular smooth muscle cells (VSMCs) [29]. Genetic interference with Axl expression impairs endothelial tube formation, which is enhanced by concomitant anti-VEGF treatment [30]. In VSMCs, Gas6/Axl-phosphoinositide 3-kinase (PI3K)-Akt signaling acts as a strong anti-apoptotic mechanism, which is important in the response to vascular injury [31]. Aggregation of platelets is dependent on TAM signaling in the presence of PtdSer residues, as shown by the Gas6/Axl-mediated tyrosine phosphorylation of integrin β3, which stabilizes thrombus formation by adhesion (Figure 1D) [32,33]. Activated platelets secrete ADP, which signals via the ADP receptor P2Y12 in cooperation with Gas6/TAMs to induce PI3K-Akt and to keep integrin αIIbβ3 in the activated state [34]. Gas6/TAM signaling promotes persistent phosphorylation of integrin αIIbβ3 and facilitates thrombus integrity, even after limited availability of ADP caused by degradation. Furthermore, Gas6/TAM receptors enhance the pro-inflammatory activation of endothelial cells by upregulating expression of VCAM-1, ICAM-1, and P-selectin, which allows binding to P-selectin glycoprotein ligand 1 expressed by leukocytes, thus promoting leukocytes extravasation and inflammation [35].

### TAM Receptors in Liver Pathophysiology

The expression of TAM receptors is post-translationally regulated by shedding of respective extracellular domains. Cleavage of Axl and MerTK close to the transmembrane region in exon 10 and exon 9, respectively, by the disintegrin and metalloproteinases (ADAM)10 and ADAM17 results in the release of soluble Axl (sAxl) and soluble MerTK (sMerTK), which can be detected in plasma and serum [24,36]. In end stage liver disease, sAxl is considered as an accurate diagnostic biomarker of liver cirrhosis and early to late stage HCC [37,38,39]. Shedding of Axl and MerTK associates with dampening of TAM signaling in melanoma or breast carcinoma as well as in liver fibrosis [40,41]. In HCC, the consequences of TAM receptor shedding on the functional impact of TAM receptors is an unresolved issue [42]. In a model of ischemic tissue injury, the expression of uncleavable Axl in bone marrow-derived macrophages revealed that Axl functions worsen cardiac repair and that cleavage of Axl is a mitigating event in this process [43]. In NASH-induced liver, the expression of cleavage-resistant MerTK in Kupffer cells fosters secretion of transforming growth factor (TGF)-β1, which activates hepatic stellate cells (HSCs) and increases NASH-mediated fibrosis, suggesting that cleavage of MerTK antagonizes fibrogenesis [40]. 

TAM receptors play a major role in chemoresistance against cytotoxic drugs and tyrosine kinase inhibitors in lung, breast, prostate, and liver carcinomas [44]. Multiple mechanisms of drug resistance exerted by Axl have been reported, including overexpression of Axl in EMT-transformed cancer cells, decreased expression of microRNAs (miRs) targeting Axl, and heterodimerization of Axl with RTKs such as epidermal growth factor (EGF)-R, Her3, MET, platelet-derived growth factor (PDGF)-R, or FGFR-3 (Figure 1E) [45,46,47,48]. In HCC, both Axl and Tyro3 are aberrantly expressed in sorafenib-resistant cancer cells, and loss-of-function studies result in increased chemosensitivity [49,50]. Similarly, augmented levels of Src homology domain-containing phosphatase 2 (SHP2) in sorafenib-resistant HCC cells correlate with RTK activation, including Axl [51]. On the contrary, a recent study revealed the upregulation of Axl after abrogation of ADAMTSL5, a crucial regulator of oncogenic signaling and chemoresistance in HCC, suggesting a role in the acquisition of chemosensitivity rather than in its escape from drug treatment [52].

## 3. Tumor-Intrinsic TAM Receptor Signaling

HCC progression frequently associates with EMT driven by transcription factors (EMT-TFs) such as Snail, Slug, Twist, and Zeb1/2 to downregulate epithelial gene expression and to concomitantly upregulate a mesenchymal-like expression program [53]. Paracrine signals from stromal cells including macrophages, fibroblasts, and myeloid-derived suppressor cells impinge on activation of EMT-TFs, resulting in EMT-transformation linked to increased migration, cancer stemness, and chemoresistance [54,55]. Recent concepts have emerged which consider EMT as a continuum comprised of intermediate hybrid states comparable with partial EMT rather than a straight forward transition to a complete mesenchymal state [55]. 

TAM receptor expression is frequently upregulated in malignant diseases as comprehensively summarized previously [20]. Axl signaling was linked to EMT in various cancer types including HCC [44,56]. Upregulation of Axl in HCC was first identified in 1998 [57], yet it took around another decade until attention was drawn towards Axl in the context of HCC progression. Axl expression is increased in invasive primary human HCC cells as well as in murine metastatic HCC cell lines [58,59]. Mechanistically, Axl is a downstream target of yes-associated protein 1 (YAP1), frizzled-2 (FZD2), STAT3, and nuclear serine/threonine kinases NUAK1/2 in HCC [60,61] (Figure 2A). Anti-tumorigenic effects by inhibition of YAP1 were partially rescued through Axl overexpression, while genetic ablation of either NUAK1 or 2 decreased Axl expression. Intervention with NUAK 1/2 and Axl reduced proliferation of HCC cells and diminished EMT-associated gene signatures [61]. In this line, interference with either Axl or Slug expression impaired the migratory potential of human HCC cell lines [62]. The clinical relevance of these findings is supported by the observation that Axl expression correlates with microvascular invasion in HCC patients [56,63]. Furthermore, Axl expression correlates with an increased risk of tumor recurrence after hepatectomy and poor patient survival [63].

Depending on the cancer entity, Axl expression is further induced by the transcription factors Sp1 and 3 as well as myeloid zinc finger 1 (Mzf1) in colorectal cancer patients (Figure 2A) [64,65]. Hypoxia-inducible factors (HIF)1/2 mediate Axl expression in different solid tumors, including renal cell carcinoma [66]. In leukemia cell lines, Axl is upregulated upon stimulation with PMA, resulting in the activation of MAPK signaling and AP-1 transcription factors, such as c-Jun [65]. In the non-malignant setting, Axl expression is induced in immune cells by Toll-like receptor (TLR) signaling, generating a negative feedback loop to control inflammatory signaling [23] (Figure 1C). In HCC, Axl activates the expression of the EMT-TFs Slug and Snail [56,62]. Importantly, Axl cooperates with TGF-β signaling as expression of Axl is crucially involved in switching TGF-β-induced growth inhibition to induction of HCC-promoting target gene expression including autocrine TGF-β signaling [56] (Figure 2B). The Axl-dependent TGF-β switch to a metastatic phenotype of HCC cells is facilitated by an altered phosphorylation pattern of the Smad3 linker region (Smad3L), thereby inducing the expression of Snail, MMP9, and Pai1. Other downstream targets of the Axl/TGF-β axis as well as their contribution to tumor progression remain to be investigated. In the murine setting, active Axl signaling downregulates expression of Cyr61, a protein involved in cell adhesion [58].

Paradoxically, a recent study showed that MerTK as well as Axl prevent TGF-β-induced EMT in alveolar epithelial cells by inducing the expression of prostaglandins [67]. Regarding HCC, the role of MerTK still needs to be investigated. Recent studies indicate EMT-promoting Tyro3 signaling in colon cancer by inducing expression of Snail [68]. Extracellular vesicles (EVs) are capable of activating TAM receptors in tumor cells, with most pronounced effects observed for Tyro3. In vitro exposure of tumor cells to EVs causes changes in cellular morphology, resulting in increased migratory potential of tumor cells—a hallmark of EMT. In this line, EV-induced Tyro3 signaling activates YAP and RhoA and promotes the expression of Slug, Snail, Twist, and N-cadherin [69] (Figure 2C). With regard to HCC, Tyro3 is suggested to favor HCC progression by positively influencing proliferation and invasion as well as by activating MAPK and PI3K/AKT signaling [49,70]. It results in increased in vivo tumor growth and serum α-fetoprotein (AFP) secretion compared to models interfering with Tryo3 activity [70]. In accordance, Tyro3 expression is increased in HCC patient samples compared to adjacent tissue and is positively correlating with tumor size and augmented levels of AFP and alanine aminotransferase [70]. Additionally, Tyro3 decreases expression of the epithelial marker E-cadherin in cellular models of human HCC [49] (Figure 2C). Tyro3 is involved in shaping an HCC-promoting inflammatory milieu by modulating the production of CXCL10 and IL8 [49]. In hepatitis-induced HCC, Tyro3 expression is induced via inflammation-dependent IL6-STAT3 signaling [71]. In this context, Tyro3 exerts its tumor-promoting functions by downstream activation of Src signaling. An autocrine feedback loop is induced, as Tyro3 in turn activates STAT3.

### TAM Receptors in Acquired Chemoresistance

EMT is a major cause of therapy resistance in cancer due to a stemness-like phenotype [61] (Figure 3). As TAM receptors are extensively linked to EMT-promoting signaling, they emerged as key players in therapy resistance. Axl signaling is a driving mechanism of acquired chemoresistance, which bypasses the targeted pathway upon drug treatment [47,61,72]. In melanoma, cancer cells initially respond to targeted therapy against MAPK; however, anti-oncogenic features are circumvented via activated Axl signaling as time on treatment increased [41]. In this line, dual treatment with Axl and MAPK inhibitors reduced tumor growth and metastasis in melanoma models [41]. An alternative mechanism of acquired chemoresistance involves the Axl-induced re-expression of argininosuccinate synthetase (Ass)1. Notably, tumor cells are dependent upon the uptake of extracellular arginine due to the loss of Ass1, an enzyme essential for de novo arginine synthesis [73]. This vulnerability is targeted by administering recombinant arginine deiminase, which degrades extracellular arginine [73]. Malignant cells, including melanoma cells, escape from treatment by reactivating expression of Ass1, which is driven by Axl-induced stabilization of c-Myc [73]. EMT-transformed non-small cell lung cancer (NSCLC) cells resistant to EGF-R treatment express high levels of Axl to bypass the blocked EGF-R, explaining their susceptibility towards pharmacological Axl inhibition [47,72]. Accordingly, dual treatment of NSCLC models with anti-EGF-R and anti-Axl therapy reduced Akt phosphorylation, cell viability, and tumor growth. The importance of the TAM receptor family in chemosensitivity is further highlighted as another member—MerTK—is linked to resistance towards anti-EGF-R and anti-NFκB treatment [74,75]. MerTK is capable of bypassing the blocked cascades, thereby rescuing cells from the anti-oncogenic treatment, and in consequence, dual therapy with either EGF-R or NFκB plus MerTK inhibitors strongly reduced viability and tumor growth [74,75].

Phospho-proteomic analysis of chemosensitive epithelial HCC cells versus intrinsically chemoresistant EMT-transformed cells revealed phospho-Axl as the predominantly activated kinase [61]. Interference with Axl signaling reduced EMT-associated kinases, while the expression of kinases correlating with an epithelial phenotype increased [61]. Mechanistically, Axl as well as NUAK 1/2 activity reduce replicational stress by downregulating DNA damage response (DDR) signaling pathways in mesenchymal HCC cells. Interestingly, Axl was linked to DDR modulation, not only in HCC [61] but also in melanoma [76], NSCLC [77,78], triple negative breast cancer (TNBC) [77], and head and neck squamous cell carcinoma (HNSCC) [77]. Pharmacological and genetic interference with Axl causes accumulation of γH2AX, a marker for double strand breaks [77,78]. Interestingly, stimulation of NSCLC, HNSCC, and TNBC cell lines with an Axl inhibitor induced specific alterations in the DDR. In NSCLC, activation of the DNA damage sensoring kinase Atr was decreased, whereas Axl inhibition in TNBC interfered with c-Myc levels and diminished levels of Atm in HNSCC [77]. Importantly, interference with Axl signaling sensitized all three indications towards PARP inhibitors [77]. In HCC, inhibition of Axl activates cell cycle checkpoint kinases Chek1/2 and cyclin dependent kinases CDK1/2, thus explaining the sensitivity of HCC cell lines to combinatory inhibition of EMT kinome (including NUAK1/2 and Axl) and check point/DDR blockade [61]. Similar outcomes were observed in NSCLC cell lines intrinsically resistant towards DDR-targeting (Atm) inhibitors, as they express higher levels of Axl compared to sensitive ones [78]. Kinase inhibitors of Axl increased phosphorylation of checkpoint kinases, while the combined treatment of NSCLC cells with Axl inhibitors and DDR-targeting Atr inhibitors significantly reduced NSCLC survival [78]. Melanoma cells intrinsically resistant to checkpoint kinase inhibitors, which are linked to DDR, can be shifted towards sensitivity by additionally targeting Axl, leading to reduced cell viability and tumor growth [76]. In conclusion, synthetic lethality induced by combinatorial treatment with Axl and DDR-targeting agents might serve as a successful strategy for increased response rates in different cancer entities, even though the events vary at a molecular level between the tumor types. 

In HCC, sorafenib-resistant SK-HEP1, an endothelial cell line originally isolated from ascetic fluid of a liver cancer patient [79], showed elevated expression of Axl [50]. A potential relevance of Axl in the resistance to HCC therapy is underlined by the second-line treatment of HCC patients with cabozantinib, a multi-kinase inhibitor targeting Axl and other RTKs [80]. Upregulation of Tyro3 was identified as a driver of acquired resistance to sorafenib in human epithelial Huh7 cells, as blocking of Tyro3 expression via miR-7 re-sensitized them again towards sorafenib [49]. In this line, Hep3B, another human epithelial HCC cell line, intrinsically expresses high levels of Tyro3 and tolerates higher doses of sorafenib [49]. Characterization of sorafenib-resistant Tyro3-expressing Huh-7 cells revealed slower proliferation as well as enhanced migratory and invasive potential. In this context, the expression of mesenchymal markers was upregulated while epithelial ones were downregulated, indicating that cells have undergone EMT [49]. The significance of TAM receptors in the context of chemoresistance is emphasized by the observation that Axl inhibition itself is bypassed by the upregulation of MerTK in models of TNBC as well as HNSCC [81].

## 4. Tumor-Extrinsic TAM Receptor Signaling

The tumor microenvironment (TME) consists of various cell types, including immune cells, endothelial cells, and fibroblasts, which direct the fate of a tumor. In HCC, the TME is composed of resident macrophages termed Kupffer cells, hepatic stellate cells (HSCs)/myofibroblasts, liver sinusoidal endothelial cells, and various immune cells such as T lymphocytes, natural killer (NK) cells, dendritic cells (DCs), neutrophil granulocytes, and peripheral macrophages. Its composition predicts disease progression and response to targeted therapy as well as immune checkpoint blockade (ICB). Upon hepatic injury, Kupffer cells are activated by injured hepatocytes in response to IL-1α and produce the pro-inflammatory cytokines TNF-α and IL-6, leading to compensatory proliferation of hepatocytes [82]. In addition, neutrophil granulocytes are simultaneously attracted to the site of liver injury to phagocytose necrotic debris and produce more pro-inflammatory cytokines. Tumor associated neutrophils, a subset of the myeloid-derived suppressor cells (MDSCs), promote an immunosuppressive environment by secreting arginase-1 (Arg1) or indoleamine 2,3-dioxygenase (IDO), therefore limiting T cell mediated killing of tumor cells [83]. DCs in the liver are responsible for type 1 IFN production and antigen presentation to activate T cells [84]. Cytotoxic CD8+ T cells kill tumor cells by secreting perforin and granzymes, subsequently inducing apoptosis in target cells [85]. Importantly, Kupffer cell-derived TGF-β activates HSC, which transdifferentiate into extracellular matrix (ECM)-producing hepatic myofibroblasts [86,87]. As a pro-fibrotic and anti-inflammatory molecule, TGF-β contributes to the inhibition of the cytotoxic activity of CD8+ T cells and NK cells, thus favoring tumor progression [88].

### 4.1. TAM and Immunity

The TAM receptors have gained attention in immunity, when MerKD mice, in which the kinase activity of MerTK is diminished, were exposed to low amounts of lipopolysaccharide (LPS), which induced a hyper-activated immune system resulting in the production of pro-inflammatory cytokines without resolution of the inflammation [89]. Similarly, TAM triple KO mice exhibit enlarged lymphoid organs, indicating hyperactivation of lymphoid immune cells by dendritic cells and macrophages [90]. Thus, the TAM receptors are crucially involved in the regulation of immune homeostasis, and their involvement in tumor immunity has gathered increasing attention [91]. Axl and MerTK have established roles in the myeloid compartment by executing efferocytosis and antigen presentation [92]. Macrophages of MerKD mice are unable to phagocytose and fail to eliminate apoptotic cells, whereas Axl and Tyro3 expression does not impact upon efferocytosis via macrophages but is crucially involved in the uptake by DCs [92]. In a model of colitis-associated colon cancer, Axl-/-MerTK-/- mice surprisingly exhibited enhanced tumor growth due to the inability of macrophages to clear the microenvironment from apoptotic neutrophils [93]. Axl-/-/MerTK-/- macrophages express higher levels of the pro-inflammatory cytokines TNF-α and IFN-γ, triggering tumor-promoting inflammation [93]. Interestingly, MerTK-/- leukocytes reduced tumor growth and expressed lower levels of anti-inflammatory cytokines such as IL-10, but higher levels of pro-inflammatory IL-6 and IL1-β in mammary cancer cells [94]. In this line, another recent study demonstrated MerTK’s tumor-promoting role in breast cancer, as its ablation increased tumor-related inflammation, therefore potentiating effects of PD-1 treatment resulting in tumor regression [95]. 

MDSCs from melanoma bearing mice express high levels of Tyro3, Axl, and MerTK, and MDSCs derived from Tyro-/-, Axl-/-, and MerTK-/-express higher levels of immunosuppressive molecules such as Arg-1, TGF-β, and IDO [96]. Additionally, the inhibition of Axl on macrophages skewed the immune microenvironment towards a pro-inflammatory and anti-tumorigenic environment, leading to better overall survival in mice, which was dependent on the cytotoxic activity of CD8+ T cells [97]. Axl, LDL receptor-related protein 1, and RAN-binding protein 9 are required for DC efferocytosis and antigen cross presentation to activate CD8+ T cells under physiological conditions and in models of viral immunity [98]. In tumor immunity, Axl expression has been found in a subgroup of DCs termed “DCs enriched in immunoregulatory molecules” (mregDCs), where it induces the expression of PD-L1 and therefore acts pro-tumorigenic [99]. NK cells are at the forefront of host defense as they manage to kill off tumor cells without requiring priming or activation in contrast to cytotoxic T cells. The TAM receptors have been identified as key regulators of NK cell functionality as they are essential for IL-15-mediated NK cell maturation [100,101]. Conversely, inhibiting the TAM receptors in NK cells interfered with proliferation and cytokine production and ultimately decreased the metastatic burden [102]. Unstimulated T cells do not express any of the TAM receptors. However, Pros1 expression is induced upon T cell receptor activation in CD4+ and CD8+ T cells dependent on IL-4 [23,103]. The expression of Axl and MerTK is quite controversial, as some studies report the expression of either one on activated T cells or no expression at all. Further investigations are required on T cell-specific TAM receptor expression to fill unmet needs and to examine their potential as targets in tumor immunotherapy.

ICB has changed the therapy landscape and has shown remarkable success in a variety of cancer entities. Currently, the two PD-1 inhibitors nivolumab and pembrolizumab are approved as ICBs in second line treatment of HCC. Notably, HCC has been clustered into three immunological subclasses based on immune cell infiltration and expression of immune checkpoint molecules, which include the (i) immune subclass (~30%), (ii) immune intermediate subclass (~45%), and (iii) immune excluded subclass (~25%) [104,105] (Figure 4A). “Hot” tumors of the immune subclass are characterized by high infiltration of immune cells, including T cells and B cells, and the upregulation of PD-1/PD-L1 expression, indicating better responses to ICB, while “cold” tumors of the immune excluded subclass are defined by decreased infiltration of cytotoxic T cells and B cells [104]. Interestingly, immune excluded “cold” HCCs showed activation of Wnt/β-catenin signaling. A small cohort study revealed that patients harboring activated Wnt/β-catenin progressed on ICB, while patients with unaltered Wnt/β-catenin signaling showed better responses [106]. Regardless of the Wnt/β-catenin status, efficacy of ICBs differs between patient subgroups based on medical history. For example, PD-1 inhibition was less effective in HCC patients with underlying NAFLD/NASH than in patients with viral hepatitis. In NAFLD accelerated HCC, intrahepatic CD4+ T cell infiltration was decreased while CD8+ T cell abundance remained unchanged [107]. Another study showed that the overall survival of NASH-driven HCC patients was reduced upon PD-1 treatment. This effect was associated with aberrantly activated exhausted CD8+ T cells that promoted tumor progression [108]. In contrast, analysis of HBV and non-viral HCC tissue revealed enrichment of PD-1+ regulatory T cells (Tregs) and CD8+ resident memory T cells, whereas exhausted Tim3+ CD8+ T cells and CD244+ NK cells were enriched in non-viral tissue. Immunosuppression of Tregs was also attenuated by PD-1 checkpoint inhibition [109]. These opposing immunophenotypes of NASH-driven HCC and HBV-HCC could explain the different responses to ICB patients based on their medical history. Accordingly, single cell analysis of treatment-naïve HCC patients revealed different subclasses of CD4+ and CD8+ T cells, including enrichment of exhausted Tregs and CD8+ T cells [110]. Profiling of human HCC tissue, human fetal liver, and mouse liver tissue showed that VEGF-A upregulates PLVAP on endothelial cells, which drives the polarization of macrophages towards an immunosuppressive phenotype [111]. Analysis of tumor and tumor adjacent tissue further revealed that CD163+ macrophages and TIGIT+ Tregs were enriched in HCC tissue, while cytotoxic CD8+ T cells and NKT cells were enriched in tumor adjacent tissue independent of HBV status [111]. In summary, increasing evidence is provided that the immune landscape of HCC leans toward a cold, immune excluded phenotype where underlying pre-dispositions impact the infiltration of immune cells and subsequently the efficacy of ICB. Therefore, future goals are to turn “cold” tumors into “hot” ones by targeting innate immunity and negative regulation of DCs, which activate cytotoxic T cells and the production of cytokines, promoting immune escape and efferocytosis, which leads to M2 polarization of macrophages [112].

In a mouse model of HBV infection in vivo, exogenous co-expression of IFN-β with HBV induced the expression of Axl in the liver [113]. DCs, expressing high levels of Axl and SOCS-1, produced IL-10, which led to differentiation of Tregs and therefore an immunosuppressive microenvironment [113]. The role of Axl during HCV infection is similar. Likewise, HCC cells upregulate the expression of Axl and SOCS3 upon HCV infection in vitro, which was linked to the production of IFN-α [114]. Overexpression of Axl dampened the expression of IFN-stimulated genes and the anti-viral response. In accordance, HCV patients harboring the IFNL3 r12979860 SNP expressed higher intrahepatic levels of Axl, showed higher viral load and responded less to anti-viral IFN treatment [114]. In a genome wide association study, a cohort of HCV infected patients carrying the rs4374383 SNP in MerTK showed higher risk of fibrosis progression [115]. The role of Tyro3 in NAFLD/NASH and HBV/HCV infection remains unclear.

### 4.2. TAM in Fibrogenesis

Continuous liver damage triggers inflammatory stimuli, which tend to become chronic once the pro-and anti-inflammatory signals are out of balance. The unresolved inflammatory milieu in the liver results in tissue remodeling, leading to accumulation of extensive amounts of ECM and formation of fibrous scars, a process considered as a pre-malignant stage [116,117]. As a result of these chronic stimuli, HSCs are activated and trans-differentiated into myofibroblasts, where they act as the main producers of ECM. Axl has been identified as a driver of liver fibrogenesis in vivo, as Axl-/- mice harbor decreased amounts of collagen in the liver compared to Axl proficient littermates [118]. Mechanistically, Gas6 activation of Axl, Akt, and the NFκB subunit p65 in HSCs leads to increased proliferation and transcription of α-SMA, TGF-β, and Col1a1. Targeting of Axl expression with bemcentinib abolished the phosphorylation of Akt and collagen deposition of HSCs [118] (Figure 4B). In accordance, Gas6-/- mice exhibit reduced fibrotic scarring in the liver compared to wildtype mice, which is accompanied by reduced infiltration of F4/80+ macrophages and inflammatory response as well as reduced levels of α-SMA and TGF-β [119]. Another study supported these findings by showing that hepatic fibrosis is only attenuated in Axl-/- mice compared to Axl-/-/MerTK-/- mice [120]. This phenotype was explained by the findings that phagocytosis of apoptotic cells in the liver and therefore decreasing inflammation is highly dependent on MerTK. Therefore, Axl-/- but not Axl-/-/MerTK-/- mice display less fibrotic livers and highlight the importance of MerTK proficient macrophages in resolving liver fibrosis [120]. 

Pre-clinical models of acute liver injury support the protective role of MerTK proficient macrophages, as MerTK-/- and Axl-/- MerTK-/- mice exhibit increased accumulation of apoptotic cells and elevated pro-inflammatory cytokines supporting liver inflammation and disease progression [120]. In this line, targeting Axl reduced or even prevented hepatic fibrosis in mice with high-fat diets, while MerTK protected primary murine hepatocytes against lipid-induced toxicity via activation of Akt (Figure 4B) [121]. Conversely, a recent study showed that macrophage-specific MerTK expression promotes liver fibrosis by upregulation of Erk-TGF-β signaling in liver macrophages, which induces activation of HSCs [40]. Clinical studies support a dichotomic function of MerTK regarding liver fibrosis, as SNP rs4374383 (within MerTK) influences fibrosis in NAFLD patients and those suffering from HCV infection [26,122]. rs4374383 AA correlates with lower MerTK expression and hepatoprotective functions, as a lower percentage of those patients suffered from stage F2–F4 fibrosis compared to patients presenting the rs4374383 GG/AG SNPs. Yet, NASH prevalence is unaffected by MerTK genotypes. Fitting with the findings from murine models, NAFLD patient samples revealed MerTK expression in HSCs and macrophages but not in hepatocytes [122]. In line with a positive regulatory role of MerTK, cleavage of MerTK by ADAM17 decreases NASH-driven progression of fibrosis. Thus, high levels of soluble MerTK delay the progression of steatosis to NASH-induced fibrosis [40]. Yet, the impact of Axl cleavage in fibrosis is an open issue. Clinical research revealed increasing levels of soluble Axl in patients with advanced fibrosis and cirrhosis compared to early stages of hepatic fibrosis, indicating a different regulatory role [38,118]. Together, these findings identify Axl as a driver of liver fibrosis by activating HSCs, whereas MerTK on macrophages acts as a pro-fibrotic in NASH-driven fibrosis and possesses an anti-fibrotic role in CCl4-driven fibrosis through apoptotic cell clearance.

### 4.3. TAM in Hemostasis and Endothelial Cells

Thrombosis is one of the major causes of cancer related mortality. The TAM receptors have been identified as essential contributors to hemostasis, as triple and double TAM knockout mice show significantly longer bleeding than wildtype mice and even single TAM receptor knockout mice [123]. TAM receptor deficiency does not prevent platelet aggregation but impairs stabilization of the platelet aggregates [32]. Gas6 and Pros1 have opposing roles in upregulating of adhesion molecules and therefore in promoting the recruitment of platelets and leukocytes. Gas6 secreted by endothelial cells upon damage stimulates TAMs, therefore acting as pro-thrombotic [124]. In contrast, Pros1 was identified as an anti-coagulating factor, as individuals with Pros1 deficiency suffer from severe recurrent venous thrombosis [125]. Pros1 acts as a co-factor of Protein C, which inactivates factor Va and VIIIa and interferes with blood clotting [126] (Figure 4C).

Axl has been identified as a pro-angiogenic molecule; thus, targeting Axl reduces tumor vascularization and tumor progression [127]. Axl expression in endothelial cells is essential for VEGF-A-dependent activation of PI3K/Akt and therefore migration of endothelial cells [128]. The role of MerTK in tumor promoting angiogenesis has not been that clear. However, miR-126 targeting MerTK suppresses breast cancer metastasis through reduced endothelial cell recruitment [129]. On the contrary, Pros1 binding to MerTK and subsequent activation of SHP2 inhibited VEGF-A mediated endothelial migration and proliferation [130].

Together, Axl and MerTK are essential regulators of physiological vascularization and tumor angiogenesis. Recently, the anti-angiogenic treatment with bevacizumab in combination with ICB has emerged as a new successful therapy regimen for patients suffering from HCC, prolonging overall survival compared to sorafenib treatment [131]. Therefore, cabozantinib targeting VEGF-R and Axl becomes an apparent treatment option. In combination with nivolumab and ipilimumab, cabozantinib has already shown promising results, as it enhances overall survival of HCC patients [132].

## 5. Conclusions and Outlook

TAM receptors emerge as promising therapeutic targets and biomarkers in chronic liver disease (CLD), although consequences of TAM activation and intervention are contrasting what might have a severe impact on clinical efforts. Axl is essentially involved in HSC activation and fibrosis, while the role of MerTK is ambiguous [40,118,120,121]. In a preclinical NASH model, treatment with the selective Axl inhibitor bemcentinib reduces NASH-associated fibrosis and concomitantly increases Gas6, suggesting a compensatory mechanism of hepatoprotective MerTK activation [121,133]. However, clinical studies in NAFLD- and HCV-induced liver fibrosis presented ambiguous functions of MerTK [26,122]. MerTK SNPs rs4374383 GG/AG correlate with higher risk and severity of liver fibrosis compared to patients harboring the rs4374383 AA genotype, indicating either pro-fibrotic or hepatoprotective functions, respectively [26,122]. The availability of high Gas6 levels after pharmacological Axl inhibition might be important in alleviating fibrosis progression. The interaction of sAxl, the level of which is significantly increased in patients with end stage CLD [38], has been proposed to reduce the availability of Gas6 and therefore have a potential hepatoprotective influence [22]. In addition, recent data suggest that inhibition of Axl phosphorylation impairs ubiquitination of Axl, resulting in increased receptor surface density and shedded sAxl [134]. On the contrary, other studies propose an excess of Gas6 and sAxl in advanced fibrosis and HCC [39,118,135]. Thus, it remains to be clarified whether complexing of sAxl with Gas6 significantly depletes the abundance of Gas6 leading to dampening of TAM receptor activation. Notably, an increase of Gas6 in the context of Axl inhibition might even fuel ongoing fibrosis, since MerTK activation in Kupffer cells causes NASH-related fibrosis [40]. In conclusion, the specific application of TAM inhibitors demands further investigation for clinical use.

Although the physiological consequences of TAM receptor shedding in CLD are still vague [42], the release of soluble receptor domains offers potential in the diagnosis and prediction of disease progression. sAxl shows comparable diagnostic accuracy to established clinical applications, such as enhanced liver fibrosis (ELF) scoring and transient elastography in the detection of liver fibrosis [38]. In addition, elevated levels of sAxl show high performance in the detection of early HCC [37,136,137]. Evaluating levels of sAxl and Gas6 from the onset of CLD might be beneficial in monitoring the progression of liver fibrosis and HCC. Yet data for Gas6 levels are less robust, as comprehensive clinical studies are lacking [135,138,139].

Increased expression of Axl in about 40% of HCC patients associates with vascular invasion and poor survival, indicating an unfavorable prognosis [56,63]. Currently, a clear molecular link between the activation of TAM receptors and primary or secondary chemoresistance is missing. Upregulation of Axl was associated with augmented levels of SHP2 in sorafenib-resistant HCC cells. Conversely, recent findings of cellular HCC models proposed that upon knockdown of ADAMTSL5, a potent mediator of oncogenic signaling and chemoresistance, Axl expression is increased [52]. Currently, a functional implication of Axl in HCC patients resistant to TKIs, such as sorafenib, remains to be elucidated [51]. The potential relevance of Axl in therapy resistance to sorafenib is indicated by second-line treatment with the Axl-inhibiting TKI cabozantinib [80]. So far, no data on Axl functions are available in the resistance against regorafenib or lenvatinib. Nevertheless, as TAM signaling is linked to EMT, which is a known driver of chemoresistance, the potential impact of TAM receptors in chemosensitivity is conceivable [53,54,55].

TAM receptors expressed by Kupffer cells and activated HSCs are upregulated during CLD progression. As the majority of HCC arises in a fibrotic background, TAM signaling in the TME might be a particularly relevant target for intervention. Additional therapeutic potential is provided by the abundance and activity of TAM receptors in tumor-controlling immune cells of the HCC microenvironment. HCC frequently displays an immune-excluded TME, which is anti-inflammatory and tumor-promoting [104,105]. Since immunotherapies such as PD-L1 inhibition show limited success in HCC [11,140,141], additional therapeutic strategies might allow one to overcome tumor/fibrosis-mediated immunosuppression. Axl and MerTK have a predominant role in the myeloid compartment of immune cells, where their activation favors an anti-inflammatory, pro-regenerative phenotype [91,92]. In line, a large body of literature reports the acquisition of an immune-evasive TME along with TAM receptor activation [94,95,96,97]. Strikingly, Axl expression is linked with PD-L1 expression [99]. Further, intervention with Axl or MerTK activation favors sensitivity to PD-1 inhibition in breast cancer and melanoma [95,96]. As recent immunotherapies in HCC focus on ICB, these findings are of particular clinical relevance. First clinical trials investigating the combination of ICB and Axl inhibition are currently ongoing in, e.g., melanoma, NSCLC, or breast cancer (NCT02872259, NCT03184571, NCT03184558). The most promising TAM-selective inhibitors currently under clinical investigation are summarized in Table 1.

Novel strategies for HCC treatment might depend on immunological subclasses and etiology. For instance, HCC patients with viral hepatitis show a better response to PD-1 inhibition when compared to patients with NASH-induced HCC [107,108]. In this line, patients of the “immune subclass” are indicated to respond better to ICB [104]. Alcohol-induced HCC reveals a poor immune cell infiltration in tumors and surrounding tissue, suggesting the rather poor efficacy of ICB [104]. Whether exaggerated TAM signaling might be attributed to a distinct subclass of HCC remains to be elucidated. In general, further efforts to stratify patients according to their immunological characteristics are essential, especially since none of the proposed classifications are clinically established, and the evaluation of tumoral PD-1/PD-L1 status alone has not emerged to be robust in HCC [11,141,142].

In HCC, different predispositions shape the immunological TME. As TAM signaling is exaggerated in pre-malignant fibrosis, the question is raised whether TAM activation shapes immunological characteristics from the early beginnings of CLD. Thus, it might be of interest whether and how augmented TAM signaling affects the TME of HCC, in particular regarding the immune landscape and EMT, its onset, and extent. Is there a predisposition of patients showing high Axl or Gas6 levels during fibrosis to develop HCC? Is it conceivable to stratify Axl^high^/Gas6^high^ CLD patients by respective serum levels for the identification of subpopulations at risk and for ensuring early detection of malignant lesions? Stratification could support the detection of patients susceptible to therapy resistance, both to TKIs and to ICB. As HCC patients are mainly diagnosed at an advanced stage with limited treatment options, the identification and monitoring of CLD patients from an early, premalignant stage might be a promising anti-cancer strategy.

## Figures and Tables

**Figure 1 cancers-13-05448-f001:**
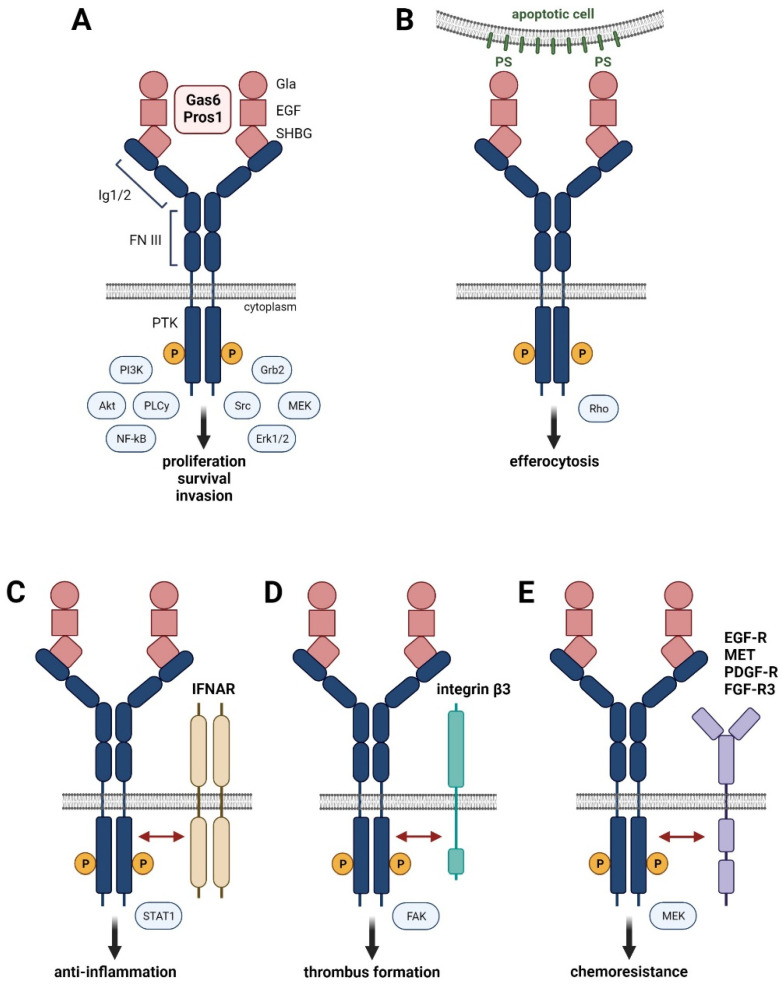
Hallmarks of Gas6-Pros1/TAM signaling. (**A**) Binding of high affinity ligands Gas6 or Pros1 to the immunoglobulin (Ig) domains of TAM receptors causes homo-dimerization and activation by phosphorylation of the intracellular protein kinase domain (PTK). (**B**) Interaction of Gas6 or Pros1 with TAM receptors and phosphatidylserine residues (PS) promotes phagocytosis of apoptotic cells. (**C**–**E**) Interaction of activated TAM receptors with interferon-α/β receptor (IFNAR) (**C**), integrin β3 (**D**), and the RTKs EGF-R, MET, PDGF-R, or FGFR-3 (**E**).

**Figure 2 cancers-13-05448-f002:**
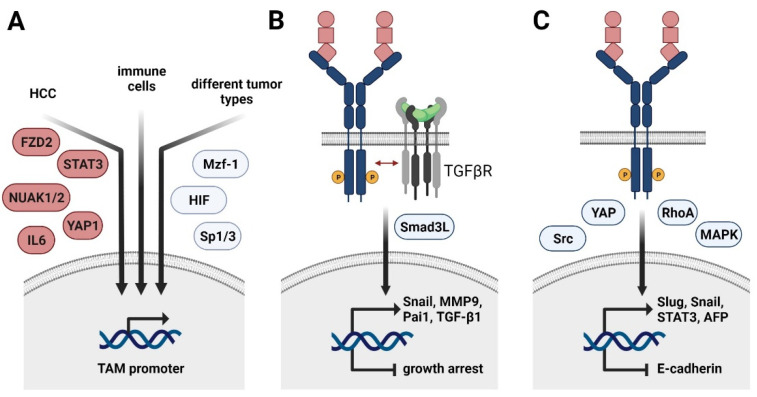
Regulation of TAM receptor expression and its role in HCC progression and EMT. (**A**) Transcriptional control of Axl expression in cancer cells of HCC (left arrow), immune cells (middle arrow), and various tumor entities (right arrow). FZD2, frizzled-2; NUAK, nuclear serine/threonine kinase; STAT3, signal transducer and activator of transcription 3, YAP1, yes-associated protein 1; Mzf1, myeloid zinc finger 1; HIF, hypoxia-inducible factor. (**B**) Molecular collaboration of Axl and TGF-β signaling in HCC switching from TGF-β-induced growth arrest to TGF-β-mediated cancer progression. Smad3L, linker region of Smad3. (**C**) HCC-specific role of TAM receptors in EMT. YAP, yes-associated protein; MAPK, mitogen-activated protein kinase; AFP, α-fetoprotein.

**Figure 3 cancers-13-05448-f003:**
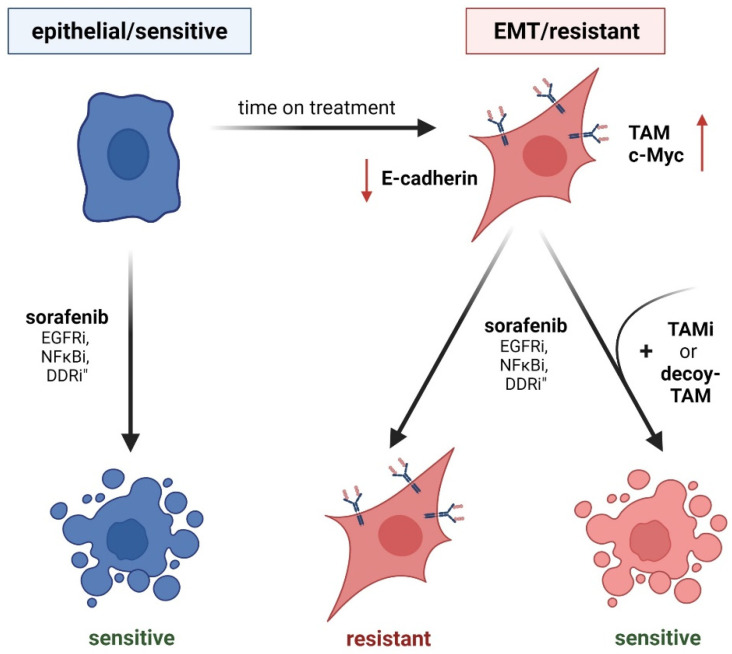
TAM receptor function in EMT and acquired chemoresistance. (Left panel) Epithelial cancer cells show sensitivity against drug treatment. EGF-Ri, inhibition of epidermal growth factor receptor; NFκBi, inhibition of nuclear factor κB; DDRi, inhibition of DNA damage response. (Right panel) EMT-transformed cancer cells having reduced levels of E-cadherin (red arrow) exhibit higher resistance to drug treatment by induced TAM expression (right panel). Inhibition of individual TAM receptors (TAMi) or treatment with soluble TAM decoy receptors that bind TAM ligands restore chemosensitivity.

**Figure 4 cancers-13-05448-f004:**
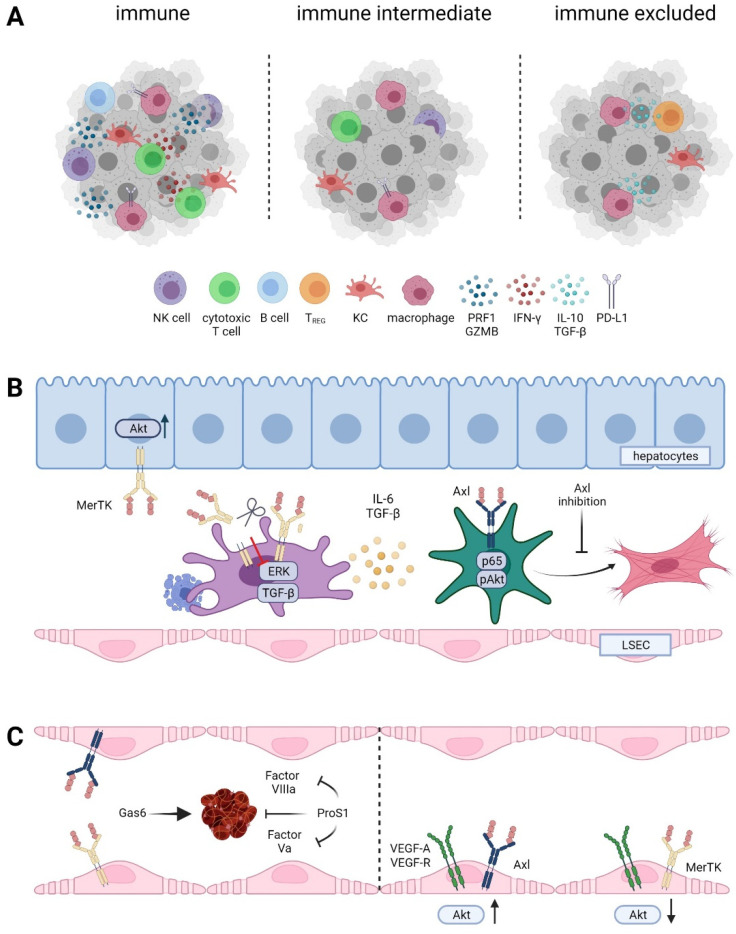
Role of TAM receptor in the tumor microenvironment of HCC. (**A**) Immune cell infiltrated “hot” versus immune intermediate versus immune cell excluded HCC classes. NK cell, natural killer cell; KC, Kupffer cell; TREG, regulatory T cell; IFN-γ, interferon-gamma; PRF, perforin; GZMB, granzyme B; PD-L1, programmed death ligand 1. (**B**) TAM receptors during fibrogenesis. LSEC, liver sinusoidal endothelial cell. (**C**) Differential role of Gas6 and Pros1 in hemostasis and vessel integrity.

**Table 1 cancers-13-05448-t001:** TAM-selective inhibition.

Drug	Targets				Cancer Type	Trial Number	Mono/Comb- Therapy	Trial Phase
	Affinity							
	Axl	MerTK	Tyro3	Additional				
Bemcentinib (BGB324 or R428)	IC_50_ = 14 nM ‡	IC_50_ = 220 nM	IC_50_ = 200 nM	RET FLT3 DDR VEGFR AURKA PDGFRα SRC	AML MDS	NCT02488408	Monotherapy +Cytarabine and Decitabine	I/II
						NCT03824080	Monotherapy	II
					Breast cancer	NCT03184558	+Pembrolizumab	II
					NSCLC	NCT03184571	+Pembrolizumab	II
						NCT02922777	+Docetaxel	I
						NCT02424617	+Erlotinib	I/II
					Mesothelioma	NCT03654833	+Pembrolizumab	II
					Pancreatic cancer	NCT03649321	+Nab-paclitaxel Gemcitabine Cisplatin	I/II
					Glioblastoma	NCT03965494	Monotherapy	I
					Melanoma	NCT02872259	+Pembrolizumab +Dabrafenib and Trametinib	I/II
Dubermatinib (TP-0903)	IC_50_ = 27 nM	nd	nd	AURKA AURKB ALK JAK	AML	NCT04518345	±Azacitidine	I/II
						NCT03013998	nd	I
					Advanced solid tumors	NCT02729298	Monotherapy	I
MRX-2843 (UNC237)	IC_50_ = 15 nM	IC_50_ = 1.3 nM	IC_50_ = 17 nM	FLT3	NSCLC	NCT04762199	+Osimertinib	I
					ALL AML MPAL	NCT04872478	Monotherapy	I
					AML	NCT04946890	Monotherapy	I/II
					Advanced solid tumors	NCT03510104	Monotherapy	I
INCB081776	IC_50_ = 0.61 nM	IC_50_ = 3.17 nM	IC_50_ = 101 nM	nd	Advanced tumors	NCT03522142	±INCMGA00012	I
ONO-7475	IC_50_ = 0.7 nM	IC_50_ = 1.0 nM	IC_50_ = 8.7 nM	PDGFRα TRKA, TRKB FLT3	AML MDS	NCT03176277	±Venetoclax	I/II
					Advanced solid tumors	NCT03730337	±ONO-4538 (Nivolumab)	I
SLC-391	IC_50_ = 9.6 nM	IC_50_ = 42.3 nM	IC_50_ = 44 nM	nd	Solid tumors	NCT03990454	Monotherapy	I
BA3011 (CAB-AXL-ADC)	nd	nd	nd	nd	NSCLC	NCT04681131	±PD-1 inhibitor	II
					Advanced solid tumors	NCT03425279	±PD-1 inhibitor	I/II
					Ovarian cancer	NCT04918186	+Durvalumab	II
Enapotamab vedotin	nd	nd	nd	nd	Solid tumors	NCT02988817	Monotherapy	I/II
AVB-S6-500 (Batiraxcept) (ultra-high affinity decoy protein)	nd	nd	nd	Gas6	Ovarian cancer	NCT04019288	+Durvalumab	I/II
						NCT03639246	+Paclitaxel or +PLD (pegylated liposomal doxorubicin)	I/II
					Ovarian cancer Fallopian tube cancer Primary peritoneal cancer	NCT04019288	+Durvalumab	I/II
					RCC	NCT04300140	+Cabozantinib	I/II
					Urothelial carcinoma	NCT04004442	+Avelumab	II
CCT 301-38 CAR-T	nd	nd	nd	nd	RCC	NCT03393936	Monotherapy	I/II

Source: www.clinicaltrials.gov, www.ema.europa.eu, and www.pubchem.ncbi.nlm.nih.gov (accessed 21 October 2021). Only active clinical trials were included. AML, acute myeloid leukemia; ALL, acute lymphoid leukemia; MDS, myelodysplastic syndrome; MPAL, mixed-phenotype acute leukemia; NSCLC, non-small cell lung cancer, RCC; renal cell carcinoma. AURKA—AURKB, aurora kinase A—aurora kinase B; ALK, anaplastic lymphoma kinase; DDR1, discoidin domain receptor family, member 1; FLT3, fms like tyrosine kinase 3; JAK, janus kinase; PDGFR, platelet derived growth factor receptor; TRKA—TRKB, tropomyosin receptor kinase A—tropomyosin receptor kinase B; VEGFR2, vascular endothelial growth factor receptor 2; nd, not determined. ‡ All values represent activity in biochemical assays.

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
