# Peer review of "Intrinsic and Extrinsic Control of Hepatocellular Carcinoma by TAM Receptors"

_cancers, 2021, doi:10.3390/cancers13215448_

Round 1

Reviewer 1 Report

In the present manuscript, Hedrich et al. summarize the current knowledge on TAM receptors in human hepatocellular carcinoma (HCC), a highly aggressive tumor with few therapeutic options. Mounting evidence indicates that the receptor tyrosine kinases Tyro3, Axl, and MerTK, which belong to the TAM family, impact multiple aspects of cancer biology, including migration, metastasis, and resistance to conventional chemotherapy and targeted drugs. In addition, the role of these proteins is becoming highly relevant also in human HCC.

Overall, this is a comprehensive, detailed, well-written, and relevant review article on a critical topic in human hepatocarcinogenesis. Thus, it will attract the attention of many readers. The review article is structured correctly; the figures are adequately designed and summarize the main messages of the article. All the relevant references on this topic are cited in the text. I have no significant concerns about the manuscript.

Minor issue:

- A small chapter and a Table summarizing the most promising TAM-selective inhibitors (although still not used in HCC) would further increase the value of the present review article.

Reviewer 2 Report

In their manuscript Hedrich and co-workers discuss recent results dealing with the role of the receptors of the TAM family Tyro3, Axl and MerTK the development of hepatocellular carcinoma (HCC) discussing the relevance of TAM receptors in modulating epithelial cell plasticity as well as immune cell infiltration and fibrogenesis.   The authors also pay attention to the role of TAM receptor in the chemoresistance against clinically used tyrosine kinase inhibitors and immune checkpoint blockade in HCC progression and on the possible use of TAM receptor inhibitors in HCC therapy.

The review is comprehensive and it is supported by extensive references.  However, the presentation of the data is not always straightforward, and it might confuse readers not deep in the field. Quite often physiological aspects of TAM receptor functions are described together their involvement in cancer biology making difficult to appreciate what extent the activity of these receptors impacts on the carcinogenic process.  I would suggest of revising the text taking into better discriminating the action of TAM receptors in physiology from those in cancer.

Reviewer 3 Report

The manuscript entitled “Intrinsic and extrinsic control of hepatocellular carcinoma by TAM receptors” aimed to review the current knowledge on the TAM receptors in the context of HCC in differentiating the effects related to TAM receptors on hepatocytes/HCC cell lines in a part named “tumor-intrinsic TAM receptor signaling”, and the effects more linked to cancer microenvironment in a part named “tumor-extrinsic TAM receptor signaling”.

In this manuscript, the authors pointed out many relevant and interesting aspects with potential therapeutic perspectives.

I consider this manuscript is well structured and documented, I only have minor comments:

- lines 206-207: a bit more details could be interesting, maybe the authors could add a synthetic table with the different TAM receptors, the malignant diseases and the references

 - Figures :

For all the figures, I recommend to add the references (numbers) related to each piece of information included in the figure

to improve the “figure 2”, I recommend to dissociate the part regarding the ”regulation” from the part regarding the ”roles”, and in the part “regulation” , to dissociate what has been shown in HCC models from what has been shown in other models.

 - lines 568-570: I disagree this conclusion (“which suggest…”) or the way it is formulated. A clarification is required.

 - English : some words and formulations sounds strange to me (even if I’m not qualified to judge about it), I suggest the English is revised by a native speaker

 - Alcohol (consumption) is mentioned in the introduction as one of the most important risk factor to develop HCC but there is no return to this etiology. I suggest adding at least one or two sentences in the conclusion part to state and explaining this fact.

Reviewer 4 Report

The review is interesting, exaustive  and well written. The topic is extremely important being focused on mechanisms involved in HCC development, diagnosis and therapeutic strategies. Only a couple of suggestions:

Pag.4, Line 126: In my opinion, you should cite, in addition, MERTK SNP able to influence liver fibrosis progression (Genome-Wide Association Study Identifies Variants Associated With Progression of Liver Fibrosis From HCV Infection, Patin et Al., GASTROENTEROLOGY 2012;143:1244–1252)

In the paragraph about TAM receptor and fibrogenesis (line 499) and/or in the conclusions and outlook(line 537) regarding the ambiguous role of MERTK in HCS activation and liver fibrosis progression, you should cite PETTA, Salvatore et Al., (2015). MERTK rs4374383 polymorphism affects the severity of fibrosis in nonalcoholic fatty liver disease. JOURNAL OF HEPATOLOGY, ISSN: 0168-8278, doi: 10.1016/j.jhep.2015.10.016.
